# Disentangling Structure and Style:
# Political Bias Detection in News by Inducing Document Hierarchy

**Jiwoo Hong**[1], **Yejin Cho**[1], **Jaemin Jung**[2], **Jiyoung Han**[2], and **James Thorne**[1]

[1]KAIST AI
[2]KAIST Moon Soul Graduate School of Future Strategy
{jiwoo_hong, yejin_cho, nettong, jiyoung.han, thorne}@kaist.ac.kr

## Abstract

We address an important gap in detecting political bias in news articles. Previous works that perform document classification can be influenced by the writing style of each news outlet, leading to overfitting and limited generalizability. Our approach overcomes this limitation by considering both the sentence-level semantics and the document-level rhetorical structure, resulting in a more robust and style-agnostic approach to detecting political bias in news articles. We introduce a novel multi-head hierarchical attention model that effectively encodes the structure of long documents through a diverse ensemble of attention heads. While journalism follows a formalized rhetorical structure, the writing style may vary by news outlet. We demonstrate that our method overcomes this domain dependency and outperforms previous approaches for robustness and accuracy. Further analysis and human evaluation demonstrate the ability of our model to capture common discourse structures in journalism.[1]

## 1 Introduction

One of the primary reasons people consume news is for social cognition: to stay informed about events and developments and make informed decisions. To fulfill this purpose, news outlets must provide citizens with information from diverse sources and perspectives. The Pew Research Center (Mitchell et al., 2018) reports that the majority of American respondents (78%) indicated that news organizations should refrain from showing favoritism towards any political party in their reporting. However, more than half of the respondents (52%) expressed dissatisfaction with the media's ability to report on political issues fairly and unbiasedly.

Extensive research has revealed that partisan bias exists in various social issues, such as the 2016 US presidential election (Benkler et al., 2018), the

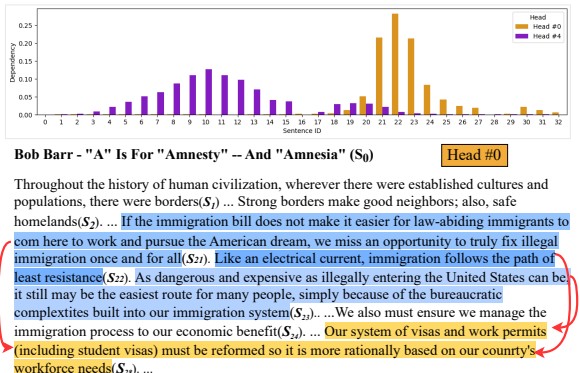

Bob Barr - "A" Is For "Amnesty" -- And "Amnesia" ($S_0$)

Throughout the history of human civilization, wherever there were established cultures and populations, there were borders($S_1$) ... Strong borders make good neighbors; also, safe homelands($S_2$). ... If the immigration bill does not make it easier for law-abiding immigrants to com here to work and pursue the American dream, we miss an opportunity to truly fix illegal immigration once and for all($S_{21}$). Like an electrical current, immigration follows the path of least resistance($S_{22}$). As dangerous and expensive as illegally entering the United States can be, it still may be the easiest route for many people, simply because of the bureaucratic complextites built into our immigration system($S_{23}$).. ...We also must ensure we manage the immigration process to our economic benefit($S_{24}$). ... Our system of visas and work permits (including student visas) must be reformed so it is more rationally based on our counrty's workforce needs($S_{28}$). ...

Figure 1: Our model identifies main sentences in an article (yellow) and the supporting sentences (blue) through the use of hierarchical multi-head attention based on their utility for the document-level classification task.

Iraq war (Luther and Miller, 2005), and climate change (Feldman et al., 2012). This disparity in media coverage has significant impacts on shaping people's perceptions of the issue (Levendusky, 2013) and their voting behavior (DellaVigna and Kaplan, 2007). Even the COVID-19 pandemic, a global health crisis, has been covered differently across the conservative and liberal political spectrum (Motta et al., 2020). Consequently, Americans have displayed a deep partisan divide, with Republicans showing less concern about personal COVID-19 risks and the severity of the pandemic than Democrats (Allcott et al., 2020). This led to Republicans being less willing to adhere to stay-at-home orders and engage in social distancing (Clinton et al., 2021), resulting in higher COVID-19 mortality rates among this group (Bendix, 2022).

The role of news media in shaping public discourse and perceptions of social issues cannot be overstated. The media plays a crucial role in disseminating information, framing issues, and setting the agenda for public debate, ultimately leading to public policy-making. Mapping the political landscape of news media is an important task. It helps news consumers evaluate the credibility of the news

---

[1]Our code is available at: https://github.com/xfactlab/emnlp2023-Document-Hierarchy

they are exposed to and more easily interpret and contextualize the information at hand. Moreover, news consumers who are aware of news outlets' political leanings can seek out other viewpoints to balance their understanding of an issue. This is particularly important in a society where media outlets are increasingly polarized along parties (Jurkowitz and Mitchell, 2020; Mitchell and Jurkowitz, 2021). It is more crucial than ever for news consumers to be discerning and critical in their news media consumption. However, previous works mainly focus on a document-level classification, making it challenging to assess *why* an article is biased.

To validate the necessity of discourse structural analysis in this task, we analyze the drawbacks of political bias classifiers that disregard the discourse structure by questioning their credibility. We formulate this issue as a *domain dependency problem*: instability of the model to which news articles model is trained or tested on. And we propose an approach for biased context detection in news articles which uses a multi-head attention mechanism to propagate document-level labels to prominent subsets of sentences, which helps make the model more reliable and explainable, illustrated in Figure 1 and Appendix G.

Our paper offers four contributions: (1) we address the problem of domain dependencies in the political bias detection task, which is an essential but understudied task; (2) we propose a new approach for biased context detection in news articles based on multi-head attention and a hierarchical model; (3) we evaluate the effectiveness of understanding the special discourse structure in the journalism domain in comparison to the syntactic hierarchy; and (4) we analyze the structures our model captures with respect to journalistic writing styles.

## 2 Background

**Measuring bias** There have been continuous efforts to determine the political bias of news outlets, with two main approaches: audience-based and content-based analyses. *Audience-based* methods assume that a news outlet's political stance can be inferred from the political preferences of its primary audience (Gramlich, 2020). This is because news outlets are expected to cater to the inclinations of their users to retain their viewership or readership. Studies have shown that partisans tend to choose news from politically congruent outlets (Davis and Dunaway, 2016; Iyengar and Hahn, 2009). However, recent web tracking technologies have revealed that a significant amount of traffic crosses ideological lines (Dubois and Blank, 2018; Flaxman et al., 2016; Gentzkow and Shapiro, 2011) This conflicting evidence suggests the audience makeup of a news outlet may no longer be stable in a highly competitive media environment.

*Content-based* methods typically rely on student coders (Feldman et al., 2012; Luther and Miller, 2005) or workers from crowdsourcing platforms (Budak et al., 2016) to measure progressive/conservative media bias. The introduction of computer-assisted content analysis techniques allows researchers to compare the linguistic patterns adopted by partisan media on a large scale (Gentzkow and Shapiro, 2010; Luther and Miller, 2005). Nonetheless, these methods are generally confined to specific news topics, such as the federal tax on inherited assets (Gentzkow and Shapiro, 2010), and may not be generalizable to other news topics or entire news channels. Recently, facial recognition algorithms have been used to quantify the visibility of politicians in broadcast news programs (Kim et al., 2022). This technique detects whether a program displays a liberal or conservative bias if it features more actors from the left (or right) for a longer duration. However, it is restricted to analyzing news visuals only and cannot be applied to other types of news content.

**Structural Properties** The writing style in the journalism domain plays a crucial role in shaping public discourse and perceptions of social issues. Sentences in the articles have unique roles according to their position in the article (Van Dijk, 1985), typically following an inverted pyramid structure (Pöttker, 2003). Structural analysis for detecting political bias in news articles (Van Dijk, 2009; Gangula et al., 2019) has shown that analysis of how information is presented can be used to determine the political bias of the author or news outlet. Therefore, specialized structural analysis of journalism is important in developing NLP-based approaches for political bias detection.

## 3 Related Works

**Detecting political ideologies and biases** There are two common approaches in the content-based political bias detection task in NLP: either article-level (Kulkarni et al., 2018; Baly et al., 2020; Liu et al., 2022), sentence-level (Chen et al., 2020; Fan et al., 2019; Spinde et al., 2021; Lei et al.,

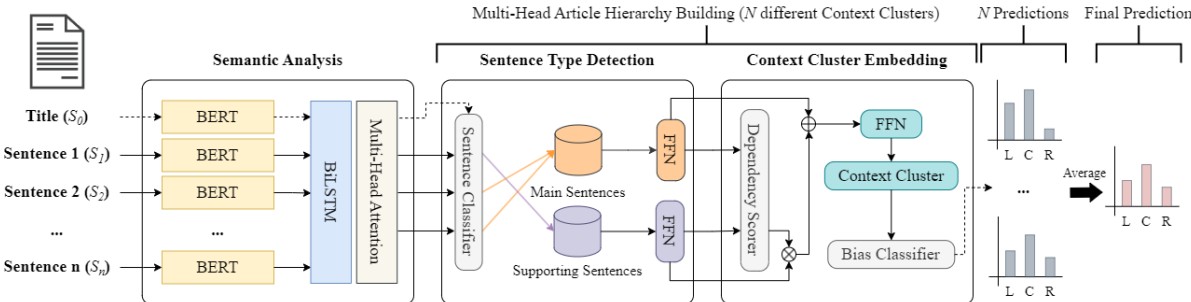

Figure 2: The model architecture. The 'Sentence Type Detection' and 'Context Cluster Embedding' sections are held independently by heads. The final prediction is an average of each head's bias prediction.

2022). For sentence-level bias detection, previous work has centered around labeling additional data: Spinde et al. (2021) developed a process for labeling data at a sentence-level with binary labels and Fan et al. (2019) defined informational bias and lexical bias while providing sentence-level and word-level bias labels for news articles. With these datasets, Lei et al. (2022) used discourse structures that inform the model about the role of a sentence to detect sentence-level bias.

Research in article-level bias centers around using external information (Baly et al., 2020; Zhang et al., 2022; Feng et al., 2022). For example, Baly et al. (2020) incorporate additional external information such as Twitter bios and Wikipedia pages related to each news outlet, and Zhang et al. (2022) and Feng et al. (2022) implemented a knowledge graph of external facts to enhance the political bias detection performance. In related works, contrastive learning with additional data preprocessing (e.g. curating article triplets) has also helped build robust models learning from annotations of left, centrist, and right viewpoints on the same event (Kim and Johnson, 2022; Liu et al., 2022).

**Hierarchical attention** Hierarchical attention (Yang et al., 2016) allows a model to encode longer texts by combining independently encoded sentences or phrases (Shen et al., 2020; Zhang et al., 2020; Han et al., 2022; Kulkarni et al., 2018; Karimi and Tang, 2019; Devatine et al., 2022). In the legal domain, Shen et al. (2020) hierarchically formulated legal events and arguments for legal event extraction. In the medical domain, Zhang et al. (2020) used hierarchical attention in medical ontology to embed medical terminologies with respect to both low- and high-level concepts.

For journalism, Karimi and Tang (2019) utilized hierarchical attention for building discourse

structure trees on fake news detection tasks. Also, for political ideology classification, Kulkarni et al. (2018) proposed a multi-view model that hierarchically encodes the article with word-level and sentence-level embeddings. Devatine et al. (2022) also hierarchically encoded the article and applied adversarial adaptation for political bias classification. While hierarchical features have been used for document-level classification, we propose propagating relevance through the document hierarchy.

**Relevance Propagation** Relevance propagation methods are widely used in many domains to improve the explainability of black box models (Nam et al., 2019; Binder et al., 2016; Bach et al., 2015). Specifically for NLP, Arras et al. (2017) propagate model weights to highlight relevant tokens for predictions. Our method widens the range of relevance propagation in NLP by considering sentence-level relevance in document understanding.

**Article-level political bias datasets** For article-level bias detection, many previous works published labeled datasets with news articles collected from Allsides.com (Kiesel et al., 2019; Baly et al., 2020; Chen et al., 2020; Liu et al., 2022). Models will likely memorize aspects irrelevant to political bias when predicting the label. Publisher (Kiesel et al., 2019), news outlet (Baly et al., 2020), and topic (Chen et al., 2020) have shown confounding effects. While Baly et al. (2020) reported this discrepancy between seen news outlets for training and unseen news outlets during testing, they did not analyze the cause of this problem.

## 4 Model Architecture

We have three modeling objectives: (1) to induce sentence-level information from a document-level label, (2) to understand the discourse structure of the news article, and (3) to reduce the effect of

spurious correlation between the media outlet and the bias label. It is critical to model the complex relation between sentences in the news article when predicting its political ideology, to this end, we apply hierarchical modeling to propagate document information to sentences.

We propose a bias prediction model that uses a two-layer tree with sentence-level embeddings to better understand the news article's semantics and discourse structure. Our pipeline consists of three key stages that can be jointly trained as a single unified model: semantic analysis, sentence type detection, and context clustering. Each component is described in detail in the following sections, and the model architecture is illustrated in Figure 2.

## 4.1 Semantic Understanding

The first stage of the model builds a representation vector for each sentence comprising the semantics, positional information, and discourse relation.

**Semantics** A news article contains a headline $S_0$ and a sequence of many sentences: $(S_1, ..., S_n)$. The semantics of each sentence is independently captured by using a large language model to generate a sentence embedding $\mathbf{s}_i = SBERT(S_i), i \in \{0, ..., n\}$. We use S-BERT (Reimers and Gurevych, 2019), as it is trained specifically to represent the semantic similarity between sentences.

**Position** The positional information captures the role of a sentence with respect to its location in the article. We chose to add the positional information to the sentence embedding through BiLSTM (Hochreiter and Schmidhuber, 1997) over encoded sentences. As the sentences in the news articles have intricate relationships more than a simple sequential relationship, we chose to use BiLSTM instead of positional encoding as we desire a conditional representation that cannot be achieved with positional encoding alone (Wang et al., 2019).

$$\mathbf{h}_i = [\overrightarrow{LSTM}(\mathbf{s}_i, \mathbf{s}_{0:i-1}); \overleftarrow{LSTM}(\mathbf{s}_i, \mathbf{s}_{n:i+1})] \quad (1)$$

**Discourse** To capture the discourse relation between multiple sentences in different viewpoints, we apply multi-head attention (Vaswani et al., 2017) over the BiLSTM encodings: $H = [\mathbf{h}_0, ..., \mathbf{h}_n]$. Each attention head, $\bar{H}_k = Attn(QW_k^{(Q)}, KW_k^{(K)}, VW_k^{(V)})$, independently models the relationships between all document sentences and headline with scaled dot product attention. We set $Q = K = V = H$. In contrast to the multi-head attention mechanism of the transformer, we do not concatenate the results. We instead perform the sentence type detection independently for each of the $N$ attention heads to propagate the political bias distribution to the independently captured $N$ different main contexts.

## 4.2 Multi-Head Sentence Type Detection

This step is computed independently for each attention head $\bar{H}_k$. For simplicity, we omit the subscript defining the head $k$. We use the attended representation to predict the importance of each of the sentences in the article with respect to the headline. Each sentence is assigned one of two roles: main or supporting. We compute these roles by comparing dot product similarity between the sentences $\bar{\mathbf{h}}_i$ and the headline $\bar{\mathbf{h}}_0$. Supporting sentences are assigned $P_{supp}(S_i|S_0) = 1 - P_{main}(S_i|S_0)$.

$$P_{main}(S_i|S_0) = \alpha_i = \frac{\exp \bar{\mathbf{h}}_i^T \bar{\mathbf{h}}_0}{\sum_{i'=1}^n \exp \bar{\mathbf{h}}_{i'}^T \bar{\mathbf{h}}_0} \quad (2)$$

We created weighted embeddings of sentences so that the sentences which are likely to be main sentences have high norms. These weighted embeddings undergo further encoding with a single feed-forward layer with GELU unit: $\mathbf{u}_i = Gelu(FFN(\alpha_i \bar{\mathbf{h}}_i))$. Similarly, we make encodings for sentences from the SUPPORTING perspective: $\mathbf{v}_i = Gelu(FFN((1 - \alpha_i)\bar{\mathbf{h}}_i))$. Note that the headline acts as a main sentence with $\alpha_0 = 1$.

## 4.3 Context Cluster Embedding

The generated perspective vectors are used for each attention head to predict the hierarchical relation between the main and supporting sentences. We use the dot product similarity to capture this discourse dependency. The dependency scorer (Figure 2) returns the proportion of focus between the main sentence $S_i$ and the supporting sentence $S_j$:

$$P_{dep}(S_j|S_i) = \frac{\exp \mathbf{v}_j^T \mathbf{u}_i}{\sum_{j'=1}^n \exp \mathbf{v}_{j'}^T \mathbf{u}_i}, i \neq j \quad (3)$$

Context cluster embeddings are created by summing the main sentence representation $u_i^{(m)}$ with the weighted sum of the supporting sentence representations. We weight this sum by the dependency score so that unrelated sentences contribute less to the context cluster embedding.

| Dataset | Media-based Split (Baly et al., 2020) | | | | | | Augmented Media-based Split (Ours) | | | | | |
|---|---|---|---|---|---|---|---|---|---|---|---|---|
| **Type** | **Train** | | **Valid** | | **Test** | | **Train** | | **Test Set 1** | | **Test Set 2** | |
| **Count** | **Data** | **Outlets** | **Data** | **Outlets** | **Data** | **Media** | **Data** | **Outlets** | **Data** | **Outlets** | **Data** | **Outlets** |
| **Left** | 8,861 | 61 | 1,640 | 14 | 402 | 7 | 7,300 | 16 | 200 | 4 | 240 | 4 |
| **Center** | 7,488 | 29 | 618 | 14 | 299 | 4 | 7,300 | 10 | 200 | 4 | 240 | 4 |
| **Right** | 10,241 | 69 | 98 | 13 | 599 | 18 | 7,300 | 8 | 200 | 4 | 240 | 4 |

Table 1: Statistics for the two datasets. The augmented version is split after merging every data in the original dataset. The news outlets in each set are mutually exclusive and randomly selected. (e.g. Daily Kos (left), CNN (left), BBC News (center), Wall Street Journal (center), Fox News (right), National Review (right))

$$\mathbf{c}_i = u_i + \sum_{j=1, j \neq i}^{n} P_{dep}(S_j|S_i) \cdot \mathbf{v}_j \qquad (4)$$

For each attention head, we create a single embedding by summing each of the context cluster embeddings $\bar{\mathbf{c}} = LayerNorm(\sum_i \mathbf{c}_i)$. We apply LayerNorm (Ba et al., 2016) to mitigate the effect of having multiple depending sentences. We then predict the distribution of bias label for the news article distribution with a linear classifier over this head's embedding: $\hat{\mathbf{y}}_k = softmax(FFN(\bar{\mathbf{c}}))$. During training, we treat each attention head as a component in a mixture model, averaging the class probabilities to predict the bias: $\hat{\mathbf{y}} = \frac{1}{N} \sum_{k=1}^{N} \hat{\mathbf{y}}_k$. At test time, we propagate the document-level label to a single main sentence and corresponding supporting sentences (i.e. context clusters) for each head according to $\arg\max_i P_{main}(S_i|S_0)$.

## 5 Experimental Settings

### 5.1 Datasets

We train and evaluate the model on the media bias detection dataset from Baly et al. (2020). Unlike Baly et al. (2020), we build two test sets to test the performance gap upon two test sets rather than pursuing the higher accuracy in a single test set. We use the `media-based` split to evaluate the generalization between news outlets to which the model is trained and tested with two modifications:[2]

First, we deleted 425 news outlets with less than 50 articles to ensure that the model is not memorizing the writing style of certain mediums. For the remaining mediums, we merged news outlets from the same company, such as "CNN" and "CNN (Web News)" to prevent overlapping between the train set and the others.

Second, we made a balanced dataset with respect to both data size and news outlets. We selected four

---
[2]Because the dataset reported in the paper significantly differs from the published files, we re-balance the data ourselves.

news outlets for each class to make our test sets. As the number of articles varied by news outlets, we selected 50 and 60 articles from each news outlet for Test Set 1 and 2, respectively. We sampled 7,300 articles from each side regardless of the news outlets to preserve the train data size for the train set. Dataset statistics and the list of news outlets for each set are provided in Table 1 and Appendix C.

### 5.2 Baseline

To compare against our hierarchical model, we also fine-tuned BERT (Devlin et al., 2019) model using the HuggingFace `bert-base-uncased` implementation (Wolf et al., 2019). We use a single classification layer over the CLS token from the final hidden layer. Since news articles are lengthy documents, the articles were truncated by 512 tokens.

## 6 Experiments

We compare our model against a BERT baseline classifier for the news article classification task with experiments to (1) compare our model against the BERT baseline, which is representative of previous work; (2) compare the accuracy of the model on two disjoint test sets (3) evaluate sensitivity to the content of training data and (4) evaluate sensitivity to the number of training data. In all experiments, we report AUROC and macro $F_1$ scores. The hyperparameters for training are in Appendix A.

**Comparison against baseline** We compare against a BERT classifier, which performs on par with the model from (Baly et al., 2020). For both the baseline and our model, we train 20 versions of the model with different random seeds. We report the average and standard deviation of the results for each model, respectively.

**Sensitivity to test data** We evaluate whether the models depend on the news articles they are tested on. We use the two test sets, which were pairwise disjoint by a news outlet (i.e. a news outlet from

| | | BERT | | Ours | | Ratio of Var. |
| | | AUROC | Macro $F_1$ | AUROC | Macro $F_1$ | (F-test) |
|---|---|---|---|---|---|---|
| **1,000** | **Test Set 1** | 0.5267 (0.0410) | 0.3633 (0.0341) | **0.6017** (0.0190) | **0.4227** (0.0185) | 4.6697*** |
| | **Test Set 2** | **0.6297** (0.0216) | **0.4639** (0.0189) | 0.5703 (0.0298) | 0.3971 (0.0250) | 0.5249(ns) |
| | **JSD (t-test)** | 0.0377*** | | **0.0289**\*\* | | - |
| **5,000** | **Test Set 1** | 0.5378 (0.0231) | 0.3831 (0.0310) | **0.6446** (0.0154) | **0.4706** (0.0172) | 2.2289* |
| | **Test Set 2** | **0.6557** (0.0250) | **0.5042** (0.0219) | 0.6507 (0.0150) | 0.4805 (0.0143) | 2.7930* |
| | **JSD (t-test)** | 0.0313*** | | **0.0163(ns)** | | - |
| **10,000** | **Test Set 1** | 0.5743 (0.0244) | 0.4123 (0.0215) | **0.6529** (0.0081) | **0.4747** (0.0090) | 9.1247*** |
| | **Test Set 2** | 0.6588 (0.0173) | 0.5079 (0.0127) | **0.6783** (0.0094) | **0.5089** (0.0122) | 3.3934** |
| | **JSD (t-test)** | 0.0165*** | | **0.0099**\*\*\* | | - |
| **Full** | **Test Set 1** | 0.5903 (0.0240) | 0.4165 (0.0146) | **0.6548** (0.0106) | **0.4751** (0.0111) | 5.1633*** |
| | **Test Set 2** | 0.6659 (0.0134) | 0.5128 (0.0107) | **0.6905** (0.0072) | **0.5247** (0.0082) | 3.4710** |
| | **JSD (t-test)** | 0.0145*** | | **0.0097**\*\*\* | | - |

Table 2: Evaluation results for BERT and ours. The first and second rows of each data size refer to the sample mean and the sample standard deviation (in the brackets) of 20 trials. The 'JSD (t-test)' row shows the JSD and the significance of the t-test. And the 'Ratio of Var. (F-test)' column shows the ratio of variance (BERT over ours) and the significance of the F-test. *, **, *** refers to $p < .05, .01, .001$ respectively, and (ns) refers to $p > .05$.

in Test Set 1 does not have any articles in Test Set 2). If the model is robust to the writing style of the news outlets in evaluation, the distribution of the tested result will be invariant between test sets. We apply a two-sided t-test over AUROC and also report divergence (JSD) between the distributions.

**Sensitivity to training data** We study if the models depend on the news articles they are trained on. We compute the learning curve by training the model on increasing subsets of data from (N=1000,5000,10000,Full). For each sample size, we report the variance based on training on 20 different subsets of the training data. We first apply the Shapiro-Wilk test (Shapiro and Wilk, 1965) to validate whether the AUROC distribution of each model on the test set is normal. Then, we conducted the F-test and checked the variance ratio to validate which model was more sensitive to the training data. We report JSD to quantify the discrepancy between the results in the two test sets for both the BERT baseline and our model.

**Learning Curve** We trained the model to the random subsets of 1,000, 5,000, and 10,000 articles and the full train set (21,900 articles) for the experiments studying sensitivity to training data.

# 7 Result

We report Macro $F_1$ and AUROC for the document-level bias prediction task in Table 2. Our results indicate that our approach (1) outperforms a conventional LM-based classifier, (2) is resilient to domain shift between train and test (3) uncovers structural properties which follow the general practice and theoretical background in journalism.

## 7.1 Comparison against baseline

To compare the data efficiency of the BERT baseline and our model, we evaluate model accuracy with varying numbers of training data in Table 2. For Test Set 1, our model outperformed BERT for both AUROC and macro $F_1$ score in every data size setting. For Test Set 2, our model outperformed BERT in the subsets of 10,000 articles and the full train set but not in the subsets of 1,000 articles and 5,000 articles. BERT outperformed when the models were trained to the small subsets and tested on Test Set 2. Otherwise, our model outperformed BERT in any setting. The highest AUROC of our model on both test sets was 0.6733 and 0.7071, outperforming BERT (AUROC of 0.6384 and 0.6977).

We further explored the reason why BERT outperformed in two specific cases with the model's sensitivity to test data in the following subsection.

## 7.2 Sensitivity to test data

To measure the discrepancy of each model's results from Test Sets 1 and 2, we used the two-sided t-test and JSD. The null hypothesis and alternative hypothesis for the t-test are listed in Appendix B.1. Except for our model trained with the subsets of

5,000 articles, it was evident to reject $H_0$. This means that both models have discrepancies in AUROC from Test Set 1 and Test Set 2. However, BERT always had a higher discrepancy measured by JSD. This indicated that BERT always showed distant results in Test Set 1 and Test Set 2 in every case. This provides evidence that our model is more robust to test data than BERT (further reported in Figure 3 and Table 2).

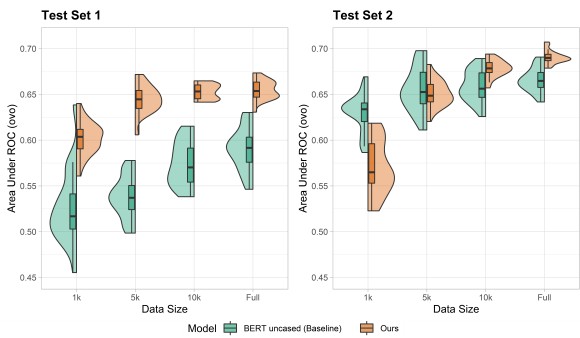

Figure 3: The learning curve of the models. Normality is assumed according to Shapiro-Wilk test ($\alpha = 0.05$).

## 7.3 Sensitivity to training data

To compare the variance of AUROC, we used the F-test. The null hypothesis $H_0$, alternative hypothesis $H_1$, and the ratio of variance $f_0$ are listed in Appendix B.2. For both models, the AUROC variance in Test Set 1 and Test Set 2 tends to decrease as the subset size increases. In all but 1 case, the variance of BERT was consistently higher than our model. Evaluating on Test Set 1, $f_0$ was always higher than $F_{0.05,19,19}$, so it was evident to reject $H_0$ in every case. Evaluating Test Set 2, $f_0$ linearly increased as the size of the subset increased. However, for 1,000 training data, we could not reject $H_0$. This shows that our model is more invariant to the data it was trained on than the BERT classifier.

## 8 Structural Analysis

In this section, we analyze the structural properties of news articles using the main sentences identified by the model. To do so, we collect the predicted main sentences and assess if they capture the formalized discourse structures commonly used in journalism. By validating from summarization and structure, we ensure the reliability of using multiple attention heads as an explanation mechanism.

We use the BASIL dataset, which contains sentence-level annotations for two types of biases (Fan et al., 2019): lexical and informational bias.

Lexical bias refers to the bias from the word choice of the journalist, such as using polarized words (e.g. *Donald Trump is investing more in conspiracy theories about President Obama's birth certificate as he explores his bid for the presidency*.) Informational bias refers to the biased elaboration of certain events or facts, which includes using selective quotations to strengthen their viewpoint. (e.g. *The Arizona group said the call from Mr. Trump on Wednesday came unexpectedly, and the group had spent much of the day Thursday scurrying to make travel arrangements to New York*.)

### 8.1 Main Sentences as Extractive Summary

We conducted machine and human evaluations to assess whether the main sentences identified by the model were informative. Our working assumption is that the concatenation of main sentences from our model acts as an extractive summary.

**Machine Evaluation** We used BART-large (Lewis et al., 2019) pretrained on CNN/Daily Mail dataset (Nallapati et al., 2016) to generate the summaries. We report BERTScore (Zhang et al., 2019) between abstractive summarizer and our model as an extractive summarizer in Table 3.

|  | Precision | Recall | F1 Score |
|---|---|---|---|
| *vs BART-Large* | | | |
| Lex. Only | 0.5741 | 0.5784 | 0.5761 |
| Info. Only | 0.7922 | 0.8144 | 0.8021 |
| Biased Sent. | 0.8253 | 0.8641 | 0.8440 |
| Our Model | **0.8739** | **0.8828** | **0.8781** |

Table 3: BERTScore between BART-Large and ours. 'Lex. Only' and 'Info. Only.' refers to the set of lexically and informationally biased sentences in the news article. And 'Biased Sent.' refers to the union of 'Lex. Only' and 'Info. Only' sentences.

**Human Evaluation** To evaluate whether the main sentences selected by the model were informative, two annotators ($\kappa = 0.43$) recorded preference of summary from either 1) the lead, 2) randomly selected sentences, 3) main sentences from our model. Without knowing the system, annotators were instructed to select which summary best explains the concept and biased contents of twenty news articles sampled from BASIL. Annotators select our model's main sentences in 75% of cases. Detailed instructions are provided in Appendix D.

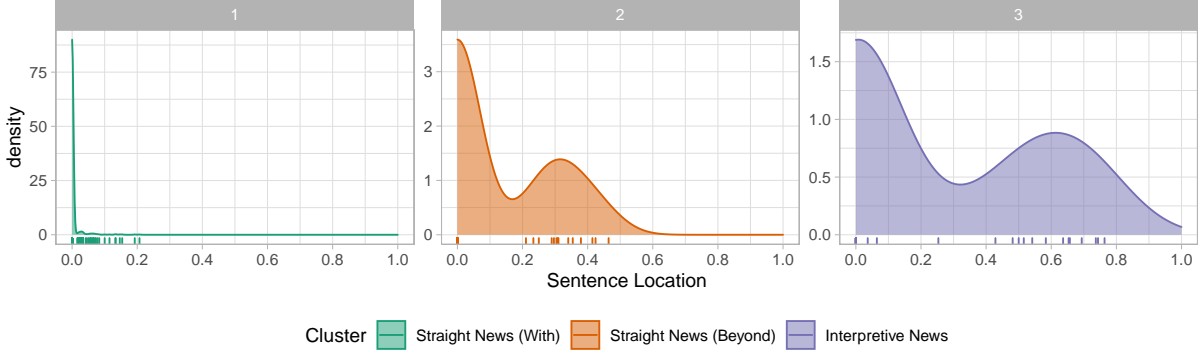

Figure 4: Density plot of the selected main sentences by clusters. The x-axis denotes the relative location of sentences, 0.0 and 1.0 indicating the first and last sentences, respectively. The rug plot below each density plot shows the actual location of the selected main sentence.

## 8.2 Structure of documents

To assess whether the main sentences identified by our model capture the structural properties of news articles, we cluster articles from the BASIL dataset with respect to their distribution of contexts using K-Means. For the distance metric, we use Dynamic Time Warping (Tormene et al., 2008, DTW), which captures the temporal elements without a dependency on the document length. The BASIL dataset consists of 100 political news stories, each comprising three articles about the same main events, sourced from the New York Times, Fox News, and the Huffington Post. Articles containing less than 200 words or exceeding 1,000 words and op-eds were excluded from the dataset.

Our analysis of the BASIL dataset, including political news longer than 200 words, identified three distinct clusters that differ in narrative structure. Clusters 1 and 2 are characterized as straight news, while cluster 3 is classified as interpretive news. The main sentence locations and statistics are listed in Table 4 and visualized in Figure 4. Sample articles are provided in Appendix F. Furthermore, we validate the clustering results in Appendix E.

| ID | Size (%) | Avg. Words | Lex. | Info. |
|---|---|---|---|---|
| **1** | **92.0** | 627.38 | 1.39 | 3.91 |
| **2** | 4.3 | 739.92 | 1.15 | 3.54 |
| **3** | 3.7 | **1107.00** | **2.27** | **6.73** |

Table 4: Statistics of clusters. Lex. and Info., refer to the average number of sentences with lexical and informational biases per article annotated in BASIL.

### 8.2.1 Cluster 1 - Straight News with the Inverted Pyramid Structure

Our classifier identified that the main sentences in cluster 1 were located in the first quarter of the article, indicating the use of the *inverted pyramid structure* (Missouri Group, 2013). In this structure, the *lead* paragraph contains the core information of the news story, while subsequent paragraphs are arranged in decreasing order of importance (Van Dijk, 1985). By presenting the essential details at the beginning, readers can rapidly comprehend the main points of the article, resulting in a shorter article (Pöttker, 2003). These characteristics of straight news using the inverted pyramid structure are reflected in the lower article length and fewer biased sentences, which is shown in Table 4.

### 8.2.2 Cluster 2 - Straight News Beyond the Inverted Pyramid

Cluster 2 comprises news articles that are of similar length to those in cluster 1 and account for 4.3% of the whole dataset. Unlike cluster 1, which follows the inverted pyramid structure, the articles in cluster 2 present the main information at the beginning and in the article's first and second quarters. This structure still employs the inverted pyramid narrative but incorporates a distinct feature – *bridge* sentences. These sentences connect anecdotes or examples to the news story's broader theme and help tie seemingly unrelated information together, leading toward a cohesive narrative thread. Interestingly, articles in cluster 2 exhibit the least amount of lexical and informational bias.

### 8.2.3 Cluster 3 - Interpretive News

Cluster 3 is distinguished by main sentences that are typically located in the first and third quarters of

the article, with the latter sentences often reflecting the reporter's interpretation of the event (Van Dijk, 1985). This type of reporting aligns with the trend toward *Interpretive news*, which seeks to uncover the meaning of news beyond the facts and statements of sources (Schudson, 1982; Patterson, 1994; Salgado and Strömbäck, 2012). As a result, interpretive news tends to be longer than straight news and often includes analysis and commentary alongside the reporting of events (Barnhurst and Mutz, 1997). Our findings are consistent with the observations made by Chen et al. (2020), who found that any political bias in news stories, if present, tends to appear in the later part of the article.

## 9 Conclusions

Our multi-head attention model leverages sentence-level information to capture the narrative structure. Our model outperforms conventional document-level classifiers by mitigating the domain dependency constraints of traditional classifiers and generating more robust and precise document-level representations. We validated our model's effectiveness in capturing journalism's rhetorical structures and writing styles.

### Limitations

While news plays a vital role in every country and language, our method only applies to English news articles. Although the dataset contains articles from some international news outlets (e.g. Al Jazeera), most news outlets are from Western countries. As political bias in news articles closely relates to cultural background, we would like to expand our work to journalism in non-English languages or based in other non-western contexts.

Our work is limited by the types of biases captured in the dataset. There are many approaches to bias, and the datasets available only reflect a limited range of discretized political bias. Future work from the community could focus on introducing better datasets and resources for modeling the many dimensions and nuances of bias.

### Acknowledgements

Jiyoung Han and James Thorne are the corresponding authors. This work was supported by the Institute of Information and Communications Technology Planning & Evaluation (IITP) grant funded by the Korean government (MSIT; Grant No. 2019-0-00075, Artificial Intelligence Graduate School Program(KAIST)), Artificial Intelligence Industrial Convergence Cluster Development project funded by the Ministry of Science and ICT(MSIT, Korea) & Gwangju Metropolitan City, and by the National Research Foundation of Korea (NRF) grant funded by MSIT (Grant No. RS-2023-00252535).

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

## Appendix

This appendix contains the following contents: (1) Hyperparameters used for training the baseline models and our models (A); (2) Notations for hypothesis test in Section 7.2 and Section 7.3 (B); (3) List of news outlets in each data set of augmented Media-based Split in Table 1 (C); (4) Human evaluation template and details used in Section 8.1 (D); (5) Sample news articles for each cluster in Section 8 (F).

## A  Hyperparameters

For our model, we used AdamW as an optimizer with a weight decay of 1e-05 for both models. Both models were trained for 25 epochs. For BERT, we used a constant learning rate of 2e-05. For our model, we used a two-layered BiLSTM with size of 512. There parameters of the S-BERT sentence encoder were frozen and not updated during training. The number of attention head was fixed to 8. And we used a 1 cycle learning rate scheduler (Smith, 2015) with a maximum learning rate of 5e-05. The maximum learning rate was reached after 10 percent of the training was finished. We used a single Nvidia RTX A6000 to train the model: BERT took 9 minutes, and our model took 22 minutes per epoch for training with the full data. Hyperparameters were selected optimizing our model for the original dataset from (Baly et al., 2020) and then applied to our augmented datasets without modification.

## B  Hypothesis Tests

This section introduces the null hypothesis $H_0$, alternative hypothesis $H_1$, and ratio of variance $f_0$ used in Section 7.2 and Section 7.3 to compare the robustness of baseline and our model.

### B.1  Sensitivity to test data

$$H_0 : \mu_{Set1} = \mu_{Set2}$$
$$H_1 : \mu_{Set1} \neq \mu_{Set2} \tag{5}$$

### B.2  Sensitivity to training data

$$H_0 : \sigma^2_{BERT} = \sigma^2_{Ours}$$
$$H_1 : \sigma^2_{BERT} > \sigma^2_{Ours} \tag{6}$$

$$f_0 = \frac{S^2_{BERT}}{S^2_{Ours}} \tag{7}$$

## C List of News Outlets in Dataset

| | |
|---|---|
| **Train** | Associated Press, Chicago Sun Times, Christian Science Monitor, BBC News, Time Magazine, Washington Times, New York Magazine, CBS News, Fox News, USA TODAY, American Spectator, Democracy Now, New York Times, MarketWatch, Reuters, Mother Jones, Pew Research Center, NPR Online News, Daily Beast, ProPublica*, Vanity Fair, CNN, BuzzFeed News, The Guardian, The Flip Side, Vox, Slate, Politico, Breitbart News, Reason, Wall Street Journal, New York Post, Newsmax |
| **Test Set 1** | The Atlantic, Yahoo! The 360, Bloomberg, Media Matters, HotAir, Salon, The Daily Caller, Victor Hanson, The Week, FiveThirtyEight, ABC News, CBN |
| **Test Set 2** | Townhall, National Review, The Hill, Business Insider, TheBlaze.com, Daily Kos, Axios, Washington Post, The Daily Wire, Vice, CNBC, Al Jazeera |

Table 5: The list of news outlets contained in the train set, Test Set 1, and Test Set 2 of the augmented media-based split dataset. The color of each news outlet is blue, black, and red if they are left-sided, centered, or right-sided respectively. (ProPublica contains both left-sided and centered news articles.)

## D Human Evaluation Template

We hired two annotators who are not experts in journalism to select the option that best follows the instructions given in Table 6. The three options were given to the annotator in a randomized order: 1) the lead of the news article, 2) set of randomly selected sentences, 3) the main sentences selected by our model. For the set of randomly selected articles, we first calculated the mean and variance of the main sentences selected by our model. Then we sampled twenty random numbers from the normal distribution with calculated mean and variance, which is used as a number of sentences to be randomly selected. After both annotations were finished, the authors measured Cohen's $\kappa$ and $F_1$ scores after counting the common and uncommon annotations.

| | |
|---|---|
| **Instruction** | These are three different sets of sentences extracted from the news article. Which of the options best explains 1) the main contents 2) the biased context of the news article? |
| **Article ID** | **Article #1** |
| **Option (1)** | *Lead of Article #1* |
| **Option (2)** | *Set of Sentences Randomly Selected from Article #1* |
| **Option (3)** | *Main Sentences that Our Model Selected from Article #1* |

Table 6: The example template of instruction and question given to the evaluators.

## E Validity of Clustering Results

We report the Silhouette score (Rousseeuw, 1987) and the additional results with changing the number of clusters. Also, we analyze the validity of Cluster 1 in Section 8.

1. **Silhouette Score:** With the three clusters reported in Table 4, the Silhouette score was 0.8988. And the score consistently decreased as we increase the number of clusters.

2. **Increasing Number of Clusters:** By clustering the news articles into three, four, and five clusters, the biggest cluster had 92% of items in $K = 3$, 91.33% in $K = 4$, and 88% in $K = 5$.

3. **Clustering within Cluster 1:** By re-clustering the news articles in Cluster 1 into minimum two to maximum five clusters, the biggest cluster had 97.6% of items in $K = 2$, 90.9% in $K = 3$, 87.3% in $K = 4$, and 86.2% in $K = 5$.

# F  Sample News Articles from Each Cluster

We sampled two articles from each cluster and show their contents. The main sentences identified by the model are italicized bold sentences. The news outlet is mentioned in the bracket next to the headline. We report BERTScore (Zhang et al., 2019) between the set of selected main sentences and a summary from a BART model (Lewis et al., 2019) pre-trained on the CNN/Daily Mail dataset (Nallapati et al., 2016).

## F.1  Cluster 1 - Inverted Pyramid Structure

| | |
|---|---|
| **Headline** | Texas Rep. Blake Farenthold resigns from Congress (Fox News) |
| **Body** | ***U.S. Representative Blake Farenthold resigned from Congress on Friday, following multiple allegations of sexual harassment, misconduct and inappropriate behavior. Farenthold, R-Texas, had previously announced that he would not seek re-election in the 2018 midterm election, when reports of sexual misconduct first surfaced.*** "While I planned on serving out the remainder of my term in Congress, I know in my heart it's time for me to move along and look for new ways to serve," Farenthold said in a statement Friday evening, noting he sent a letter to Texas Gov. Greg Abbott to tell him about his resignation, effective at 5 p.m. "It's been an honor and privilege to serve the constituents of Texas' 27th Congressional District. I would like to thank my staff both in Washington and Texas for all of their hard work on behalf of our constituents. I would also like to thank my family for their unwavering support and most importantly the people that elected me," Farenthold said. "Leaving my service in the House, I'm able to look back at the entirety of my career in public office and say that it was well worthwhile." Farenthold was sued by his former aide, Lauren Greene, in 2014, alleging a hostile work environment, gender discrimination and retaliation. Among other things, Greene claimed Farenthold asked her for a threesome. She sued him and was paid $84,000 from a public fund on behalf of Farenthold for a sexual harassment claim. Farenthold pledged to repay the $84,000 in taxpayer money spent to settle the claim. Farenthold's former communications director, Michael Rekola, also described in detail the congressman's alleged abusive behavior toward staff members, which ranged from making sexually graphic jokes to verbally abusing his aides. Other staffers accused Farenthold of routinely commenting on the size of women's breasts and making jokes about being on "redhead patrol" because he was attracted to women with red hair. |
| **BERTScore** | **0.9337** |

Table 7: Example of an inverted pyramid structure article from cluster 1.

| | |
|---|---|
| **Headline** | Jesse Jackson Jr. Pleads Guilty in Campaign Money Case ([New York Times](#)) |
| **Body** | *WASHINGTON — Jesse L. Jackson Jr., the former Democratic representative from Illinois, pleaded guilty on Wednesday to one felony fraud count in connection with his use of $750,000 in campaign money to pay for living expenses and buy items like stuffed animals, elk heads and fur capes. As part of a plea agreement, prosecutors recommended that Mr. Jackson receive a sentence of 46 to 57 months in prison.* The federal judge overseeing the case, Robert L. Wilkins, is scheduled to sentence Mr. Jackson on June 28. "For years I lived off my campaign," Mr. Jackson, 47, said in response to questions from the judge about the plea. "I used money I shouldn't have used for personal purposes." At one point during the hearing, the judge stopped his questioning of Mr. Jackson, who was crying, so that he could be given a tissue. "Guilty, Your Honor — I misled the American people," Mr. Jackson said when asked whether he would accept the plea deal. Mr. Jackson's father, the Rev. Jesse L. Jackson, his mother and several brothers and sisters accompanied him to the hearing. Mr. Jackson's wife, Sandi, also accompanied him, and later in the day she pleaded guilty to a charge that she filed false income tax statements during the time that Mr. Jackson was dipping into his campaign treasury. Prosecutors said they would seek to have her sentenced to 18 to 24 months. Mr. Jackson's plea was yet another chapter in the downward spiral of his career. Elected to Congress in 1995 at the age of 30 from a district that includes part of the South Side of Chicago, Mr. Jackson was once one of the most prominent young black politicians in the country, working on issues related to health care and education for the poor. But as the federal authorities investigated Gov. Rod R. Blagojevich of Illinois over his efforts to sell the Senate seat that President Obama vacated in 2008, they uncovered evidence that one of Mr. Jackson's friends had offered to make a contribution to Mr. Blagojevich's campaign in exchange for the seat. Since then, Mr. Jackson, who has said he had no knowledge of the offer, has been dogged by questions about his ethics. Last summer, Mr. Jackson took a medical leave from Congress and was later treated for bipolar disorder. After winning re-election in November, he resigned, citing his health and the federal investigation into his use of campaign money. After the hearing, Mr. Jackson's lawyer, Reid H. Weingarten, said his client had "come to terms with his misconduct." Mr. Weingarten said that Mr. Jackson had serious health issues that "directly related" to his conduct. "That's not an excuse, it's just a fact," Mr. Weingarten said. Court papers released by federal prosecutors on Wednesday provided new details about how Mr. Jackson and his wife used the $750,000 in campaign money to finance their lavish lifestyle. From 2007 to 2011, Mr. Jackson bought $10,977.74 worth of televisions, DVD players and DVDs at Best Buy, according to the documents. In 2008, Mr. Jackson used the money for things like a $466.30 dinner at CityZen in the Mandarin Oriental in Washington and a $5,587.75 vacation at the Martha's Vineyard Holistic Retreat, the document said. On at least two instances, Mr. Jackson and his wife used campaign money at Build-A-Bear Workshop, a store where patrons can create stuffed animals. From December 2007 through December 2008, the Jacksons spent $313.89 on "stuffed animals and accessories for stuffed animals" from Build-A-Bear, according to the documents. One of the more exotic items they bought was an elk head from a taxidermist in Montana. According to the documents, Mr. Jackson arranged in March 2011 to have $7,000 paid to the taxidermist, with much of the money coming from a campaign account, and it was shipped a month later to Mr. Jackson's Congressional office. A year later, Mr. Jackson's wife, knowing that the elk head had been bought with campaign money, had it moved from Washington to Chicago, and she asked a Congressional staff member to sell it, the documents say. In August 2012, the staff member sold the elk head for $5,300 to an interior designer and had the money wired to one of Mr. Jackson's accounts. What the staff member did not know was that the interior designer was actually an undercover F.B.I. employee who was investigating the Jacksons, the documents say. Documents released on Friday showed how Mr. Jackson used his campaign money to buy items like fur capes, celebrity memorabilia and expensive furniture. Among those items were a $5,000 football signed by American presidents and two hats that once belonged to Michael Jackson, including a $4,600 fedora. |
| **BERTScore** | **0.9195** |

Table 8: Example of an inverted pyramid structure article from cluster 1.

## F.2   Cluster 2 - Straight Reporting

| | |
|---|---|
| **Headline** | Republicans challenge Clinton claims on budget cuts, Benghazi cable (Fox News) |
| **Body** | *Republicans are challenging a host of statements made by Secretary of State Hillary Clinton and Democratic allies during Wednesday's heated Libya testimony – claiming that complaints about a lack of funding are bogus and questioning the secretary's insistence she never saw urgent cables warning about the danger of an attack.* The questions come as the Senate Foreign Relations Committee begins its confirmation hearing for Sen. John Kerry, D-Mass., who was tapped to replace Clinton at the department. One issue that may come up is the department's funding. Assertions that State Department posts are left vulnerable because Congress has decided not to fully fund security requests pervaded Wednesday's hearings. "Shame on the House for ... failing to adequately fund the administration's request," Rep. Gregory Meeks, D-N.Y., said Democratic New York Rep. Eliot Engel repeatedly said Congress had "slashed" diplomatic security requests. ***Clinton, in turn, affirmed their claims, saying budget issues are a "bipartisan problem."*** Budget numbers, though, actually show the overall diplomatic security budget has ballooned over the past decade. Democrats point to modest decreases in funding in recent years, and the fact that Congress has approved less than was requested. But Congress often scales back the administration's requests, and not just for the State Department. And the complaints tend to overlook the fact that the overall security budget has more than doubled since fiscal 2004. For that year, the budget was $640 million. It steadily climbed to $1.6 billion in fiscal 2010. It dipped to $1.5 billion the following year and roughly $1.35 billion in fiscal 2012 – still far more than it was a decade ago. Slightly more has been requested for fiscal 2013. It's difficult to tell how much was specifically allocated for Benghazi. Tripoli was the only post mentioned in the department's fiscal 2013 request – funding for that location did slip, from $11.5 million in fiscal 2011 to $10.1 million the following year. Slightly more has been requested for fiscal 2013. Still, then-Deputy Assistant Secretary for Diplomatic Security Charlene Lamb testified in October that the size of the attack – and not the money – was the issue. Asked if there was any budget consideration that led her not to increase the security force, she said: "No." She added: "This was an unprecedented attack in size." Asked again about budget issues, Lamb said: "Sir, if it's a volatile situation, we will move assets to cover that." Asked Wednesday about Lamb's testimony, Clinton noted that the review board that examined the Libya attack found budget issues have played a role. "That's why you have an independent group like an (Accountability Review Board); that's why it was created to look at everything," Clinton said. But Rep. Dana Rohrabacher, R-Calif., said "any suggestion that this is a budget issue is off base, or political." Other lawmakers further complained that the State Department has spent millions on lower-priority projects that could have been spent on security. Another pivotal issue Wednesday dealt with an Aug. 16 cable. That cable summarized an emergency meeting the day before by the U.S. Mission in Benghazi and warned the consulate could not defend against a "coordinated attack." That cable is seen as one of the vital warnings sent out of Libya in the months leading up to the attack. But, to the dismay of lawmakers, Clinton repeatedly said she never saw it. "That cable did not come to my attention. I have made it very clear that the security cables did not come to my attention or above the assistant secretary level," Clinton said. "I'm not aware of anyone within my office, within the secretary's office, having seen the cable." Rep. Michael McCaul, R-Texas, said "somebody within your office should have seen this cable." "An emergency meeting was held and a cable sent out on Aug. 16 by the ambassador himself, warning of what could happen. And this meant this cable went unnoticed by your office. That's the bottom line," he said. Clinton said it was "very disappointing" that "inadequacies" were found in the "responsiveness of our team here in Washington," and said "it's something we're fixing and intend to put into place protocols and systems to make sure it doesn't happen again." The secretary tried to explain that "1.43 million cables" come through the department every year. They are addressed to her but in many cases do not go to her. Rather, they go through "the bureaucracy." Republicans argue the Aug. 16 cable was rather high priority. As Sen. Rand Paul, R-Ky., put it, "Libya has to have been one of the hottest of hot spots around the world." He claimed that not knowing about their security requests "really, I think, cost these people their lives." "Had I been president at the time, and I found that you did not read the cables from Benghazi, you did not read the cables from Ambassador Stevens, I would have relieved you of your post. I think it's inexcusable," Paul said. |
| **BERTScore** | **0.8849** |

Table 9: Example of straight reporting article from cluster 2.

| | |
|---|---|
| **Headline** | 2020 Candidates Demand Full and Immediate Release of Mueller Report ([New York Times](#)) |
| **Body** | *CHARLESTON, S.C. — Democratic presidential candidates wasted no time Friday evening demanding the immediate public release of the long-awaited report from Robert S. Mueller III, with several saying that Americans deserved to know any findings about President Trump, Russia and the 2016 election in order to form judgments about Mr. Trump and the 2020 race.* Former Representative Beto O'Rourke, campaigning in South Carolina on Friday night, told reporters that "those facts, that truth, needs to be laid out for all Americans to be able to make informed decisions going forward, whether at the ballot box" or in their discussions with their senators and representatives. Mr. O'Rourke, asked if he supported impeaching Mr. Trump, said he believed the president and his 2016 campaign "at least sought to collude with the Russian government to undermine our democracy" and that Mr. Trump "sought to obstruct justice" once in office. "I think those are grounds enough for members of the House to bring up the issue of impeachment," he said. "But whether they do or not, this will ultimately be decided by the American public at the ballot box in South Carolina and in every state of the union." No voters brought up the Mueller report to Mr. O'Rourke as he spoke to them Friday. *Indeed, in the hours after the announcement of the report, Democratic candidates reacted over Twitter or in remarks at events rather than in back-and-forth conversations with voters.* Several candidates, in calling for the swift release of the report, also sought to gather new supporters and their email addresses by putting out "petitions" calling for complete transparency from the Justice Department. "The Trump Administration shouldn't get to lock up Robert Mueller's report and throw away the key," Senator Cory Booker argued on Twitter, asking people to sign a petition and provide their names and emails. Such information is often used for future fund-raising solicitations. Within hours of Mr. Mueller's completing his investigation, Senator Elizabeth Warren and the campaign arm of House Democrats were already placing ads on Facebook demanding the full report's release, with the Democratic Congressional Campaign Committee seeking 100,000 signatures for its online petition. With no detailed information available about the report, Ms. Warren and Mr. Booker — as well as Senators Kirsten Gillibrand and Kamala Harris — sought to focus attention and pressure on how quickly Attorney General William P. Barr would release the full report. "Attorney General Barr — release the Mueller report to the American public. Now," Ms. Warren wrote on Twitter. Ms. Gillibrand made similar demands and also retweeted the news of the report along with three words: "See you Sunday." That is when Ms. Gillibrand plans to formally kick off her 2020 campaign in front of Trump International Tower in New York. Ms. Harris, in addition to calling for the report to be released "immediately," called on Mr. Barr to "publicly testify under oath about the investigation and its findings." Ms. Harris, Ms. Warren and Ms. Gillibrand also joined Mr. Booker and Senator Bernie Sanders in asking supporters to sign their petitions calling for the report's immediate release. Four other candidates — Senators Amy Klobuchar of Minnesota; Gov. Jay Inslee of Washington; former Representative John Delaney; and Julián Castro — also called for the release of the full report. "As Donald Trump said, 'Let it come out,'" Mr. Sanders wrote on Twitter. "I call on the Trump administration to make Special Counsel Mueller's full report public as soon as possible. No one, including the president, is above the law." |
| **BERTScore** | **0.8585** |

Table 10: Example of straight reporting article from cluster 2.

## F.3 Cluster 3 - Interpretive Reporting

| | |
|---|---|
| **Headline** | Supreme Court Questions Claims in Sex Bias Suit Against Wal-Mart (Fox News) |
| **Body** | *A pending class action lawsuit against Wal-Mart that would be the largest of its kind in U.S. history may soon be dismissed, considering the tenor of oral arguments before the Supreme Court Tuesday.* Although she may ultimately side with the plaintiffs, even Justice Ruth Bader Ginsburg, an ardent defender of women's rights, expressed some concerns about the particulars of the sex discrimination lawsuit that covers 1.5 million women and could cost the world's largest retailer billions of dollars. The key vote for a Wal-Mart victory could belong to Justice Anthony Kennedy, who said he was troubled by what he called an inconsistency in the women's lawsuit. "Number one, you said this is a culture where Arkansas knows, the headquarters knows, everything that's going on," Kennedy told lawyer Joseph Sellers who represents the women. "Then in the next breath, you say, well, now these (local) supervisors have too much discretion." The lawsuit alleges that the company's corporate culture, described in court Tuesday as the "Wal-Mart Way," fosters the advancement of male workers over their female counterparts. It also claims that despite a company policy expressly prohibiting discrimination, local store managers are given too much flexibility in determining salary hikes and job promotions that invariably favor men over women. This dual argument that confounded Justice Kennedy also drew the ire of Justice Antonin Scalia who said he was whipsawed by the claims. "If somebody tells you how to exercise discretion, you don't have discretion," he said. Chief Justice John Roberts also offered his doubts about the merits of lawsuit, suggesting that any discriminatory acts at Wal-Mart are no worse than anywhere else. "Is it true that Wal-Mart's pay disparity across the company was less than the national average?" he asked. Sellers said that wasn't a fair comparison because Wal-Mart has an obligation under federal law to make sure its managers do not discriminate. The case started a decade ago when Wal-Mart worker Betty Dukes said the management at her Pittsburg, Calif., store was bypassing her for promotions. "I could see the men going forth and the women in the store stayed in the basic positions they were always in," Dukes once told an interviewer. *Her discrimination claims were folded into a class action lawsuit covering all current female Wal-Mart employees and any who worked for the company going back to late 1998.* Two lower courts said the case could move toward trial. Wal-Mart's appeal is asking the Supreme Court to stop the case from ever getting to a trial judge. On Tuesday, Dukes walked out of the courthouse full of confidence and poise saying she feels no anger toward her bosses or anyone else. "Wal-Mart may be a big company and that is no doubt. But they are not big enough where they can't be challenged in a court of law. If you do wrong, then you should be held accountable. From the least of us to the greatest of us." *Dukes's case has drawn the attention of the larger business community who fear that if the justices allow the case to proceed, it will open the doors to more class action lawsuits.* The U.S. Chamber of Commerce and major corporations including Bank of America, General Electric and Microsoft submitted briefs in the case supporting Wal-Mart. Much of the hour-long argument delved into the tedious details of class action law and if the Wal-Mart women could properly certify their claims into a single case. Wal-Mart lawyer Theodore Boutrous argued that every member of the class couldn't possibly meet a standard of commonality to justify the lawsuit. "Our expert's report and testimony showed that at 90 percent of the stores, there was no pay disparity," Boutrous told the court. "And that's the kind of – and even putting that aside, the plaintiffs needed to come forward with something that showed that there was this miraculous recurrence at every decision across every store of stereotyping, and the evidence simply doesn't show that." Another technical concern that appears to work against the women covers the different types of remedies they are seeking. In addition to the punitive damages they want an injunction that would force Wal-Mart to adopt more stringent anti-discrimination policies. But those two remedies require different standards for class certification and Justice Ruth Bader Ginsburg said it was "a very serious problem" in the case to try and sue for punitive damages after only obtaining certification under the lower threshold required for the injunction. It's possible that instead of an outright victory for Wal-Mart, the justices could issue a split decision of sorts and allow only the injunction part of the lawsuit to move forward. That potential ruling would get Wal-Mart off the hook for any financial damages. |
| **BERTScore** | **0.8650** |

Table 11: Example of interpretive reporting article from cluster 3.

* This sample article is divided into three pages.

| | |
|---|---|
| **Headline** | Trump Declares a National Emergency, and Provokes a Constitutional Clash (New York Times) |
| **Body(Cont.)** | *WASHINGTON — President Trump declared a national emergency on the border with Mexico on Friday in order to access billions of dollars that Congress refused to give him to build a wall there, transforming a highly charged policy dispute into a confrontation over the separation of powers outlined in the Constitution.* Trying to regain momentum after losing a grinding two-month battle with lawmakers over funding the wall, Mr. Trump asserted that the flow of drugs, criminals and illegal immigrants from Mexico constituted a profound threat to national security that justified unilateral action. "We're going to confront the national security crisis on our southern border, and we're going to do it one way or the other," he said in a televised statement in the Rose Garden barely 13 hours after Congress passed a spending measure without the money he had sought. "It's an invasion," he added. "We have an invasion of drugs and criminals coming into our country." But with illegal border crossings already down and critics accusing him of manufacturing a crisis, he may have undercut his own argument that the border situation was so urgent that it required emergency action. "I didn't need to do this, but I'd rather do it much faster," he said. "I just want to get it done faster, that's all." The president's decision incited instant condemnation from Democrats, who called it an unconstitutional abuse of his authority and vowed to try to overturn it with the support of Republicans who also objected to the move. "This is plainly a power grab by a disappointed president, who has gone outside the bounds of the law to try to get what he failed to achieve in the constitutional legislative process," Speaker Nancy Pelosi of California and Senator Chuck Schumer of New York, the Democratic leader, said in a joint statement. Mr. Trump's announcement came during a freewheeling, 50-minute appearance in which he ping-ponged from topic to topic, touching on the economy, China trade talks and his coming summit meeting with North Korea's leader, Kim Jong-un. The president again suggested that he should win the Nobel Peace Prize, and he reviewed which conservative commentators had been supportive of him, while dismissing Ann Coulter, who has not. Sounding alternately defensive and aggrieved, Mr. Trump explained his failure to secure wall funding during his first two years in office when Republicans controlled both houses of Congress by saying, "I was a little new to the job." He blamed "certain people, a particular one, for not having pushed this faster," a clear reference to former Speaker Paul D. Ryan of Wisconsin, a Republican. Mr. Trump's assertions were replete with misinformation and, when challenged by reporters, he refused to accept statistics produced by his own government that conflicted with his narrative. "The numbers that you gave are wrong," he told one reporter. "It's a fake question." On point after point, the president insisted that he would be proved correct. "People said, 'Trump is crazy,'" he said at one point, discussing his outreach to Mr. Kim. "And you know what it ended up being? A very good relationship." Mr. Trump acknowledged that his declaration of a national emergency would be litigated in the courts and even predicted a rough road for his side. "Look, I expect to be sued," he said, launching into a mocking riff about how he anticipated lower court rulings against him. "And we'll win in the Supreme Court," he predicted. Indeed, Public Citizen, an advocacy group, filed suit by the end of the day on behalf of three Texas landowners whose property might be taken for a barrier. California and New York likewise announced that they will sue over what Gov. Gavin Newsom of California called the president's "vanity project," and a roster of other groups lined up to do the same. "Fortunately, Donald Trump is not the last word," said Mr. Newsom, a Democrat. "The courts will be the last word." Among those predicting a flurry of judicial decisions against Mr. Trump was George T. Conway III, a conservative lawyer and the husband of Kellyanne Conway, the president's counselor. "If he knows he is going to lose," Mr. Conway, a vocal critic of Mr. Trump, wrote on Twitter, "then he knows he is violating the Constitution and laws he has sworn to uphold." The House Judiciary Committee announced Friday that it would investigate the president's emergency claim, while House Democrats plan to introduce legislation to block it. That measure could pass both houses of Congress if it wins the votes of the half-dozen Republican senators who have criticized the declaration, forcing Mr. Trump to issue the first veto of his presidency. The emergency declaration, according to White House officials, enables the president to divert $3.6 billion from military construction projects to the wall. **(Continued)** |

**Body(Cont.)**  Mr. Trump will also use more traditional presidential discretion to tap $2.5 billion from counternarcotics programs and $600 million from a Treasury Department asset forfeiture fund. Combined with $1.375 billion authorized for fencing in the spending package passed on Thursday night, Mr. Trump would have about $8 billion in all for barriers, significantly more than the $5.7 billion he unsuccessfully demanded from Congress. The president opted not to tap hurricane relief money from Texas or Puerto Rico, an idea that had generated angry complaints from Republicans. But he expressed no concern that diverting military construction money would delay projects benefiting the troops like base housing, schools and gyms. "It didn't sound too important to me," he said. Neither the White House nor the Pentagon had yet identified which projects may be shelved as a result, but Pentagon lawyers and other officials planned to work over the weekend to identify which construction funds would be diverted. ***The declaration also provided that land may be transferred to the Defense Department from other federal agencies or from privately purchased or condemned land.*** The next step would be to secure those lands, where the Pentagon would erect barriers. The declaration gives Patrick Shanahan, the acting defense secretary, broad latitude to carry out this process. Most Americans oppose Mr. Trump's emergency declaration, according to polls. One released this week by Fox News found 56 percent against it, including 20 percent of Republicans. Mr. Trump's desire for approval by Fox and other conservative news outlets was on display when he identified various pundits as supporters, naming Sean Hannity, Laura Ingraham, Tucker Carlson and Rush Limbaugh, although he insisted that "they don't decide policy." But Ms. Coulter, who has viscerally attacked Mr. Trump for caving on the wall, has clearly gotten under his skin. "I don't know her," he said before quickly correcting himself. "I hardly know her. I haven't spoken to her in way over a year." He noted, though, that she was an early predictor of his election victory. "So I like her, but she's off the reservation," he said. "But anybody that knows her understands that." Ms. Coulter fired back shortly afterward. "The only national emergency is that our president is an idiot," she said on KABC radio in Los Angeles. White House officials rejected criticism from across the ideological spectrum that Mr. Trump was creating a precedent that future presidents could use to ignore the will of Congress. Republicans have expressed concern that a Democratic commander in chief could cite Mr. Trump's move to declare a national emergency over gun violence or climate change without legislation from Congress. "It actually creates zero precedent," Mick Mulvaney, the acting White House chief of staff, told reporters. "This is authority given to the president in law already. It's not as if he just didn't get what he wanted so he's waving a magic wand and taking a bunch of money." ***Presidents have declared national emergencies under a 1970s-era law about five dozen times, and 31 of those prior emergencies remain active.*** But most of them dealt with foreign crises and involved freezing property, blocking trade or exports or taking other actions against national adversaries, not redirecting money without explicit congressional authorization. White House officials cited only two times that such emergency declarations were used by presidents to spend money without legislative approval — once by President George Bush in 1990 during the run-up to the Persian Gulf war, and again by his son, President George W. Bush, in 2001 after the terrorist attacks in New York, Washington and Pennsylvania. In both of those cases, the presidents were responding to new events — the Iraqi invasion of Kuwait and Al Qaeda's assault on the United States — and were moving military funds around for a military purposes. Neither was taking action specifically rejected by Congress. In Mr. Trump's case, he is defining a longstanding problem at the border as an emergency even though border apprehensions have actually fallen in recent years, to 400,000 in the last fiscal year from a peak of 1.6 million in the 2000 fiscal year. And unlike either of the Bushes, he is taking action after failing to persuade lawmakers to go along with his plans through the regular appropriations process. The spending package passed Thursday by Congress included none of the $5.7 billion that Mr. Trump demanded for 234 miles of steel wall. Instead, it provided $1.375 billion for about 55 miles of fencing. Mr. Trump signed the package into law on Friday anyway to avoid a second government shutdown after the impasse over border wall funding closed the doors of many federal agencies for 35 days and left 800,000 workers without pay. For weeks, Republicans led by Senator Mitch McConnell of Kentucky urged Mr. Trump not to declare a national emergency, but the president opted to go ahead anyway to find a way out of the political corner he had put himself in with the failed effort to force Congress to finance the wall. Mr. McConnell privately told the president that he would support the move despite his own reservations, but warned Mr. Trump that he had about two weeks to win over critical Republicans to avoid having Congress vote to reject the declaration. Mr. Trump was among those Republicans who criticized President Barack Obama for using his executive authority to spare millions of illegal immigrants from deportation after failing to persuade Congress to do so. **(Continued)**

| | |
|---|---|
| **Body (Cont.)** | "Repubs must not allow Pres Obama to subvert the Constitution of the US for his own benefit & because he is unable to negotiate w/ Congress," Mr. Trump tweeted in 2014. But Mr. Trump sought to drive home the personal toll of illegal immigration, inviting to the Rose Garden several relatives of Americans killed by people in the country without authorization. Some of the relatives, known as "angel moms," stood up holding pictures of loved ones who had died. "Matthew's death was preventable and should have been prevented," one of the women, Maureen Maloney, said in an interview after the event. Her son Matthew Denice, 23, was killed in 2011 in a motorcycle accident in Massachusetts after colliding with an automobile driven by an undocumented immigrant. "He should have never been here in the first place," she said. "If he wasn't here, it wouldn't have happened." |
| **BERTScore** | **0.8576** |

Table 12: Example of interpretive reporting article from cluster 3.

# G   Example Model Output

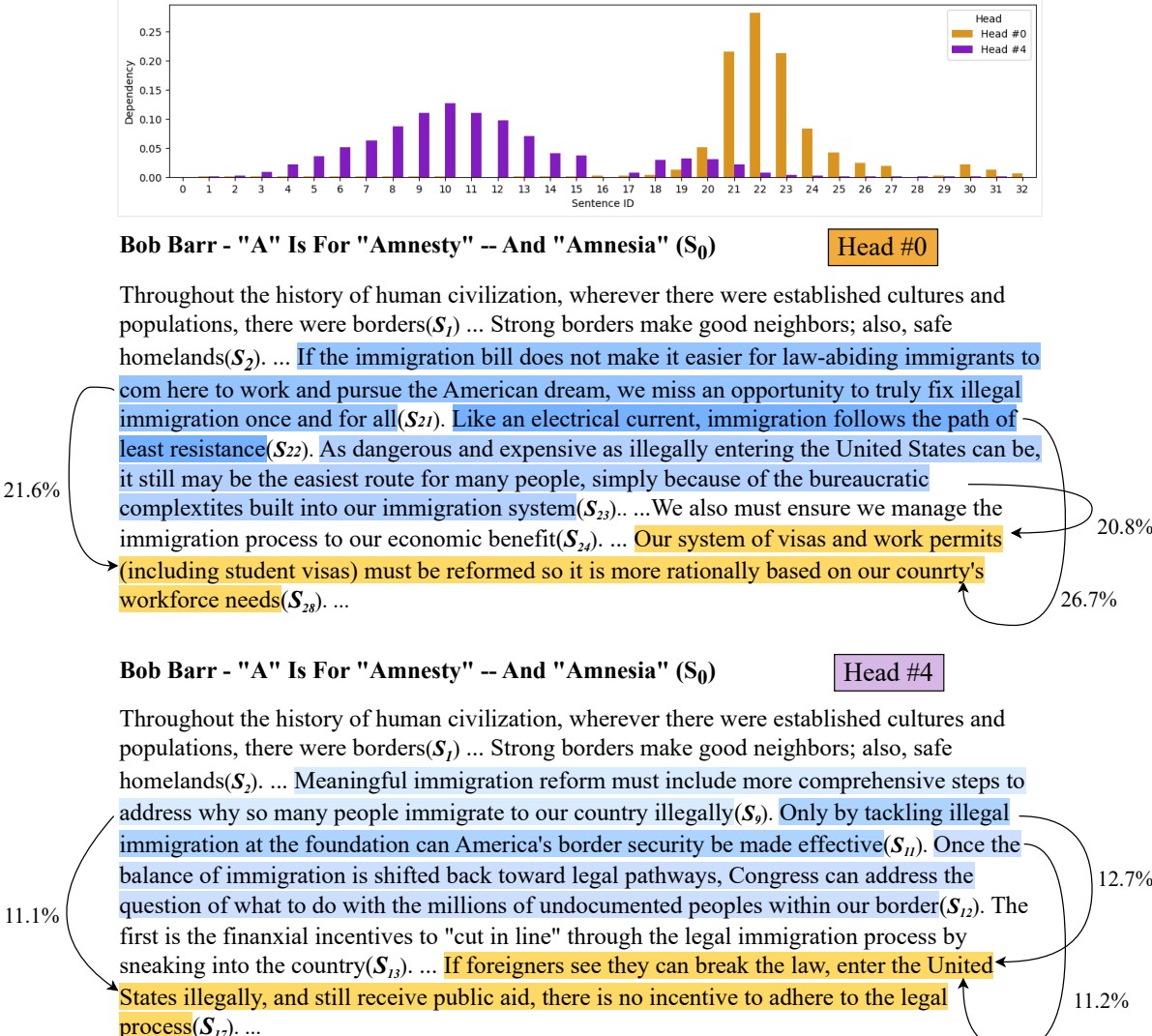

**Bob Barr - "A" Is For "Amnesty" -- And "Amnesia" ($S_0$)**    Head #0

Throughout the history of human civilization, wherever there were established cultures and populations, there were borders($S_1$) ... Strong borders make good neighbors; also, safe homelands($S_2$). ... If the immigration bill does not make it easier for law-abiding immigrants to com here to work and pursue the American dream, we miss an opportunity to truly fix illegal immigration once and for all($S_{21}$). Like an electrical current, immigration follows the path of least resistance($S_{22}$). As dangerous and expensive as illegally entering the United States can be, it still may be the easiest route for many people, simply because of the bureaucratic complextites built into our immigration system($S_{23}$).. ...We also must ensure we manage the immigration process to our economic benefit($S_{24}$). ... Our system of visas and work permits (including student visas) must be reformed so it is more rationally based on our counrty's workforce needs($S_{28}$). ...

21.6%   20.8%   26.7%

**Bob Barr - "A" Is For "Amnesty" -- And "Amnesia" ($S_0$)**    Head #4

Throughout the history of human civilization, wherever there were established cultures and populations, there were borders($S_1$) ... Strong borders make good neighbors; also, safe homelands($S_2$). ... Meaningful immigration reform must include more comprehensive steps to address why so many people immigrate to our country illegally($S_9$). Only by tackling illegal immigration at the foundation can America's border security be made effective($S_{11}$). Once the balance of immigration is shifted back toward legal pathways, Congress can address the question of what to do with the millions of undocumented peoples within our border($S_{12}$). The first is the finanxial incentives to "cut in line" through the legal immigration process by sneaking into the country($S_{13}$). ... If foreigners see they can break the law, enter the United States illegally, and still receive public aid, there is no incentive to adhere to the legal process($S_{17}$). ...

11.1%   12.7%   11.2%

Figure 5: Our model identifies main sentences in an article (yellow) and the supporting sentences (blue) through the use of hierarchical multi-head attention based on their utility for the document-level classification task.