# OpenReview forum: "Disentangling Structure and Style: Political Bias Detection in News by Inducing Document Hierarchy"
_EMNLP/2023/Conference — EMNLP 2023 Findings_

### Official Review · Reviewer_twSB · 2023-07-30

**Soundness:** 3

**Excitement:**

2: Mediocre: This paper makes marginal contributions (vs non-contemporaneous work), so I would rather not see it in the conference.

**Missing References:**

The literature gap is large in the paper. The authors should have cited and compared their model's performance with other existing models. A few of them are listed below.

- Encoding social information with graph convolutional networks for political perspective detection in news media. (Li and Goldwasser, 2019)
- Using social and linguistic information to adapt pretrained representations for political perspective identification. (Li and Goldwasser, 2021)
- CLoSE: Contrastive Learning of Subframe Embeddings for Political Bias Classification of News Media. (Kim and Johnson, 2022)
- KCD: Knowledge Walks and Textual Cues Enhanced Political Perspective Detection in News Media. (Zhang et al., 2022)
- KGAP: Knowledge Graph Augmented Political Perspective Detection in News Media. (Feng et al, 2022)

Please have a look at the papers citing these papers and more models for this task might be found.

**Paper Topic And Main Contributions:**

The authors of this paper studied political bias detection in news articles. For that, the authors proposed a model that takes into account both sentence-level semantics and document-level discourse structure of the articles for bias detection. The authors compared their model's performance against BERT and performed a set of qualitative evaluations of their model's prediction. The authors performed structural analysis of the news articles centered around -- in what part the main sentence appears in the news articles. And they found that the findings using their proposed model correspond to previous findings, indicating the correctness of their approach.

**Reasons To Accept:**

[A1] The proposed model for bias detection seems novel and easy to replicate which allows it to be used by researchers in future studies.

[A2] The experiments done in the paper seems sound and the qualitative experiments are interesting.

**Reasons To Reject:**

[R1] Bias detection is a well-studied task at this point and the authors sidelined a lot of works and models previously proposed for bias detection (refer to missing citations). As a result, the authors compared their model's performance only with BERT, which is not a state-of-the-art model when it comes to political bias detection in news articles. Because BERT is not a good encoder when it comes to encoding long contexts such as news articles. Different approaches have been proposed to perform political bias detection in news articles that are directly applicable to the setting of this paper. Without comparison with those models, it is difficult to understand the effectiveness of the proposed approach in this paper.

[R2] The proposed model even underperforms BERT in the low-training set and balanced test data setting (refer to Table 2). However, I did not find a good justification for this low performance in the paper. Does it mean the proposed model requires more training data and has some bias toward a particular class label? Per-class classification scores would be really helpful to evaluate that.

**Reproducibility:**

5: Could easily reproduce the results.

**Reviewer Confidence:**

4: Quite sure. I tried to check the important points carefully. It's unlikely, though conceivable, that I missed something that should affect my ratings.

---

> ### Author Rebuttal · Authors · 2023-08-29
>
> We would like to thank you for the helpful reviews and detailed recommendations for related works that you gave. We are happy to provide additional experiments and analysis on both the existing and additionally reported experimental results. Additionally, we've included a brief overview of our objectives and purpose. We trust this will enhance the clarity of our manuscript.
>
> **[R1: Comparison with Other Models]**
>
> Thank you for highlighting the importance of a comprehensive comparison with models other than BERT. We strongly concur that considering models that are specialized for political bias detection, can be helpful for fair comparison. During our research, however, we had difficulty finding relevant models suitable for comparison.  While there are specialized models for political bias detection like POLITICS (Liu et al., 2022) it is impossible to test them in our dataset setting as they require additional information or pre-processing from the plain text of the news articles: external knowledge (Baly et la., 2020, Zhang et al., 2022, Feng et al., 2022), specifically knowledge triplets (Liu et al., 2022, Kim and Johnson et al, 2022). In contrast, our method emphasizes content-based classification without the use of knowledge augmentation, ensuring easy reproducibility and independence from external knowledge sources. We view it as a significant strength of our model and for its distinctness, it is difficult to find models with a similar methodology for a direct comparison.
>
> Though we couldn't find directly comparable state-of-the-art models for a fair comparison, we tried to compare the performance of our model with that of the recently proposed political bias detection models which uses external knowledge. However, it is worth noting that most works in the realm of political bias detection, even those recommended by the reviewer, refrain from releasing their code and dataset for public access (e.g. Li and Golwasser et al., 2019 and Li and Golwasser et al., 2021 did not open code, Zhang et al., 2022 did not open data pre-processing code). Given this context, we chose to showcase our model's superior performance within the same dataset configuration as other models. Subsequently, our aim was to underscore that even robust language models can be affected by the composition of their training and test sets. Through these endeavors, our objective was to emphasize the significant dependence of political bias detection models on the specific news articles used in their training and testing phases. In that sense, we report the results on GPT2 as a language model with a longer input token limit and BERT-Large as a stronger encoder model compared to the BERT-base.
>
> **[R2: Baseline Outperforming Proposed Models]**
>
>  We were also aware of the phenomena you’ve mentioned, and we would like to provide further explanation on the low performance in low-training set setting. The underlying motivation of the paper was that capability of the model which can be shown through accuracy, F1 score, or AUROC is no longer credible if it is highly variant according to the training and test sets. In that sense, BERT as a political bias classifier is relatively harder to rely as Jansen-Shannon Distance is consistently higher than our proposed method. As our model have more parameters compared to BERT, it is natural for our model to be more variant in low-training set setting. However, our model is robust even in low-training set setting. We believe our joint analysis on model’s performance metric (e.g. AUROC, F1 Score) and robustness can question the performance of BERT even though it shows higher performance in a few low-training set cases.
>
> To add, we report the experiments done with BERT-Large and GPT2. We experimented BERT-Large to empirically show that the variance of AUROC and F1 Score should be higher as the number of model’s parameters increase. Also, we experimented GPT2 to empirically show that same phenomena that we focus on also happens in stronger language model with more parameters and more input token limits. We believe these experimental results will further support the insights given in Section 7 of our paper.
>
>
>
> |        |            | GPT2                |                     | BERT-LARGE      |                 | Ours                |                     |
> |--------|------------|---------------------|---------------------|-----------------|-----------------|---------------------|---------------------|
> |        |            |        AUROC        |       Macro F1      |      AUROC      | Macro F1        | AUROC               | Macro F1            |
> | 1,000  | Test Set 1 | 0.5050 (0.0260)     | 0.3214 (0.0145)     | 0.5229 (0.0279) | 0.3180 (0.0585) | **0.6017 (0.0190)** | **0.4227 (0.0185)** |
> |        | Test Set 2 | **0.6185 (0.0174)** | **0.4353 (0.0428)** | 0.5858 (0.0532) | 0.3871 (0.0849) | 0.5703 (0.0298)     | 0.3971 (0.0250)     |
> | 10,000 | Test Set 1 | 0.5756 (0.0122)     | 0.3883 (0.0425)     | 0.5596 (0.0158) | 0.3841 (0.0131) | **0.6529 (0.0081)** | **0.4747 (0.0090)** |
> |        | Test Set 2 | 0.6564 (0.0134)     | 0.4833 (0.0269)     | 0.6559 (0.0231) | 0.4912 (0.0183) | **0.6783 (0.0094)** | **0.5089 (0.0122)** |
>
> **[Overview]**
>
> *Q. How does our research differ from prior studies in this field?*
>
> - We contribute a new modeling formulation that examines and addresses the domain dependency issue in political bias detection models, particularly the challenge of "Predicting political bias for unfamiliar news sources that arise post-training."
>
>
> - Earlier models for detecting political bias required external resources, such as dataset-adjusted triplets (Liu et al., 2022, Kim and Johnson et al, 2022) or external knowledge (Baly et la., 2020, Zhang et al., 2022, Feng et al., 2022). In contrast, our model leverages multi-head attention and a hierarchical structure, relying solely on the article and its structure. This ensures simplicity in reproduction and eliminates dependence on external knowledge sources or specific datasets that make reproduction challenging.
>
>
> - We also incorporated journalism-specific features in discourse structure to verify our model's alignment with authentic knowledge. Specifically, our contributions in section 8 examine whether our model's structural analysis aligns with journalistic writing styles of straight reporting and opinion pieces. This is an emergent property of our model helping to validate our findings.
>
> *Q. Why did we opt for our dataset formulation?*
>
> - The Baly et al., 2020 dataset we use posed certain challenges which we mitigate. The dataset available on GitHub contains a contamination of news articles between the train, validation and test splits. This makes it difficult to analyze a model on the resilience to unseen writing styles.
> - Recognizing this, we re-partitioned the dataset so that it has two distinct test sets that pairwise disjoint between the test and training sets. We created test set 1 and 2. Each of these sets, along with the training set, contains a mutually exclusive set of news outlets with perfectly balanced labels. This addition not only the contamination issue of Baly et al., 2020, it also rectifies a further imbalance issue and aids in a more comprehensive assessment of our model's performance across varied news sources.
>
> - To aid reproducibility of our work, we will release our data splits as well as our code

---

### Official Review · Reviewer_b9Zn · 2023-08-03

**Soundness:** 3

**Excitement:**

2: Mediocre: This paper makes marginal contributions (vs non-contemporaneous work), so I would rather not see it in the conference.

**Missing References:**

None I know

**Paper Topic And Main Contributions:**

This paper describes a method for political stance prediction using rhetorical structure, which avoids problems of source prediction by style matching.   They propose a new model of heirarchical analysis with multi-headed attention, and apply this model to cluster articles by rhetorical structure.

**Questions For The Authors:**

(1) If you repeat the the k-means analysis on just the large class 1 docuemnts, do you get a meaningful partition of this class into subclasses?   (Delete class 2 and 3 documents before doing this analysis)

**Reasons To Accept:**

(1) Their approach avoids the problem of recognizing writing styles of sources instead of principled analysis of bias, which seems an interesting idea

(2) The machine learning models and statistical significance analysis are sophisticated

(3) there is an effort to identify distinct article structures using k-means clustering

**Reasons To Reject:**


(1) among the multiple papers I have refereed on stance prediction in this cycle, this is the one that I somehow found most difficult to understand.   See my comments on writing below.

(2) The analysis of structure type of Section 8 was not really convincing to me.   The k-means analysis showed that 92% of the articles reflected the inverted pyramid. with the other two clusters so small as to seem insignificant.

(3) There are some weird results where the baseline model out-performed their model for smaller test sizes, although I applaud the authors for the honesty of including the non-flattering data in the paper.

**Reproducibility:**

3: Could reproduce the results with some difficulty. The settings of parameters are underspecified or subjectively determined; the training/evaluation data are not widely available.

**Reviewer Confidence:**

3: Pretty sure, but there's a chance I missed something. Although I have a good feel for this area in general, I did not carefully check the paper's details, e.g., the math, experimental design, or novelty.

**Typos Grammar Style And Presentation Improvements:**

The paper seems well written on the surface, yet I had more difficulty understanding things than other papers on the topic which I received.  Part of it may be the assumption of certain terms which need to be defined.    for example, the domain dependency problem is defined in terms of an "instability" of the model.

Similarly, the reasons for the augmented media-based splits were not clear to me.  My sense is that you had good reasons, but the explanations of motivation were not really adequate.

---

> ### Author Rebuttal · Authors · 2023-08-29
>
> We appreciate your feedbacks on our paper and suggestions that can improve the clarity of our paper. We would like to provide the responses to your points starting with ‘Reasons to Reject’ section, and continue with ‘Question to the Authors’ section.
>
> **[R1: Clarity of Writing]**
>
> Thank you for the feedback. We will work to make sure the contents of the paper are accessible for a wider range of readers from practitioners in journalism as well as NLP.  We will provide a brief response to the two observations you made regarding the clarity of our work and we hope that this will provide clearer insights into our contributions.
>
> *Q. How does our research differ from prior studies in this field?*
>
> - We contribute a new modeling formulation that examines and addresses the domain dependency issue in political bias detection models, particularly the challenge of "Predicting political bias for unfamiliar news sources that arise post-training."
>
>
> - Earlier models for detecting political bias required external resources, such as dataset-adjusted triplets (Liu et al., 2022, Kim and Johnson et al, 2022) or external knowledge (Baly et la., 2020, Zhang et al., 2022, Feng et al., 2022). In contrast, our model leverages multi-head attention and a hierarchical structure, relying solely on the article and its structure. This ensures simplicity in reproduction and eliminates dependence on external knowledge sources or specific datasets that make reproduction challenging.
>
>
> - We also incorporated journalism-specific features in discourse structure to verify our model's alignment with authentic knowledge. Specifically, our contributions in section 8 examine whether our model's structural analysis aligns with journalistic writing styles of straight reporting and opinion pieces. This is an emergent property of our model helping to validate our findings.
>
> *Q. Why did we opt for our dataset formulation?*
>
> - The Baly et al., 2020 dataset we use posed certain challenges which we mitigate. The dataset available on GitHub contains a contamination of news articles between the train, validation and test splits. This makes it difficult to analyze a model on the resilience to unseen writing styles.
> - Recognizing this, we re-partitioned the dataset so that it has two distinct test sets that pairwise disjoint between the test and training sets. We created test set 1 and 2. Each of these sets, along with the training set, contains a mutually exclusive set of news outlets with perfectly balanced labels. This addition not only the contamination issue of Baly et al., 2020, it also rectifies a further imbalance issue and aids in a more comprehensive assessment of our model's performance across varied news sources.
>
> - To aid reproducibility of our work, we will release our data splits as well as our code
>
> **[R2: Validity of Structural Analysis]**
>
> We also agree that the size of Cluster 1 is dominant. Also, we deeply understand that it might look like an insignificant result without an explanation of the specialty in the journalism domain. In addition to the results we reported in Q1, we report the Silhouette score as a metric for within-cluster similarity and inter-cluster dissimilarity, and the clustering results with different numbers of clusters.
>
> **1) Silhouette Score:** Silhouette score in three cluster settings was 0.8988 where the score near 1 indicates the strong similarity within the clusters and strong dissimilarity between the clusters.
>
> **2) The Cluster Size by Differing the Number of Clusters:** We report the size of the biggest cluster by differing the number of clusters from three to five. The biggest clusters in each setting took 92% (result in our paper, K=3),  91.33% (K=4), and 88% (K=5) of the whole set of news articles in the BASIL dataset.
>
>
> **[R3: Baseline Outperforming Proposed Model]**
>
> The motivation and the most interesting point that we found in political bias detection as a downstream task in the field of NLP were that they were critical to both the training and test sets. So, we wanted to empirically validate the presence of the observed phenomena, making it essential to conduct experiments across various settings. In the scenario involving 1,000 instances, BERT might show higher accuracy in Test Set 1, but its performance significantly fell in Test Set 2. We believe the accuracy of the model is reliable if it is invariant to both the training and test sets. In that sense, we would like to note that the learning curve of BERT is distinct on Test Set 1 case and Test Set 2 case as shown in Figure 3.
> 	Furthermore, we report the results of GPT2 and BERT-Large trained and tested to our dataset to show that domain dependency problem occurs even in the models with more parameters and more input token limits.
>
> **[Q1: Re-Clustering in Cluster 1]**
>
> Along with the response to R2, we report the results of clustering on top of the news articles in Cluster 1 to answer your question. We conducted clustering with a minimum of two to a maximum of five different clusters, and the biggest clusters took 97.6% (K=2), 90.9% (K=3), 87.3% (K=4), and 86.2% (K=5) of news articles in original Cluster 1.

---

### Official Review · Reviewer_Nv3C · 2023-08-06

**Soundness:** 3
**Typos Grammar Style And Presentation Improvements:** Line 198

**Ethical Concerns:**

Yes

**Excitement:**

4: Strong: This paper deepens the understanding of some phenomenon or lowers the barriers to an existing research direction.

**Justification For Ethical Concerns:**

No ethical consideration section found in this paper. Considering this paper is tackling the political bias detection problem which is full of sensitive data and information, there is a pressing need to discuss some ethical perspectives (e.g., copyright of news data, failure mode of your bias detection model, its impact to vulnerable populations).

**Missing References:**

Late Fusion with Triplet Margin Objective for Multimodal Ideology Prediction and Analysis (echo line 142, this work also attempts to detect political bias using visuals/images)

**Paper Topic And Main Contributions:**

The author study document-level bias detection, in the format of predicting among Left, Center and Right on the basis of US political spectrum. To this end, the author proposes a multi-stage model in which the model first capture semantic meaning of each sentence, then predict the sentence type (i.e., main sentence v.s. supporting sentence), and finally aggregate representations for document-level bias detection by computing the dependency scores between each pair of sentence. The author further conducts an unsupervised-style study where the document-level label is propagated to sentence-level in order to identify main and supporting sentences. Through this propagation, the author is able to uncover some interesting writing patterns commonly employed in journalism, e.g., reversed pyramid.

**Questions For The Authors:**

For semantic analysis component (4.1), did you try to use a standard transformer model in place of your "BiLSTM + Multi-Head Attn" since the transformer model has positional embedding in nature and implements the attn mechanism? If your concern is the concatenation of multi-heads results, you could manually decompose it.

Line 452: In order to use figure 3 to draw the conclusion that your model is more robust, would it be better to have a another figure in which you could directly show the performance difference between test set 1 and 2, and you could still put your green region and orange region side-by-side. This way, readers will be more easily to visualize the sensitivity to test data of different models.

In table 3, what are the meanings of "Lex only", "Info only", "biased sentence". It's strongly suggested to provide the meaning of each baseline either in text or in captions.

**Reasons To Accept:**

This paper is well written and easy to follow (except for section 4; see my questions below). The author also presents thorough related work.

The author conducts extensive experiments, and employ several statistical tools to validate their results and findings. Most of their experimental results seem to be statistically sound (see my concern of section 7.3 conclusion in "Reasons To Reject").

The experimental results are insightful. For example, interesting structural analysis are presented in section 8, which uncovers structural properties which follows the general practice and theoretical background in journalism. The unsupervised learning approach used here is also interesting, and opens up new possible directions for detecting sentence-level bias.

**Reasons To Reject:**

Comparison is not comprehensive. One of your cited work, POLITICS (liu et al, 2022), developed a new LM trained on news articles with a special training objective. To draw a fairer comparison, models more powerful than BERT need to be included.

Section 7.3: I don't think rejecting the null hypothesis could validate the claim "This shows that our model is more invariant to the data it was trained on than the BERT classifier". Instead, rejecting the null hypotheses in almost all but 1 case shows that *your model is more invariant to the random seeds*. In order to show that your model is invariant to the training data, your null hypothesis should be something like "mu_1000 = mu_5000 = mu_10000 = mu_full", i.e., multiple mean difference comparison. To this end, you may consult ANOVA.

Despite the interestingness of section 8, I am concerned about the validity. In other words, if you run your k-means algorithm again, you may end up with a different cluster? It's advised to run k-means multiple times and try to determine the cluster id of an article based on its "aggregated score".

Some writings in section 4 are not clear. For example, you should bold u_i in equation 4, and you are advised to explicitly say that "the propagation is for structual analysis" in line 332-335, otherwise people might get confused why you want to propagate the label to sentence-level.

**Reproducibility:**

4: Could mostly reproduce the results, but there may be some variation because of sample variance or minor variations in their interpretation of the protocol or method.

**Reviewer Confidence:**

4: Quite sure. I tried to check the important points carefully. It's unlikely, though conceivable, that I missed something that should affect my ratings.

---

> ### Author Rebuttal · Authors · 2023-08-29
>
> First of all, we appreciate your detailed reviews and kind suggestions for the overall content of our paper. With additional statistical analysis and experimental results, we provide our response to the ‘Questions For the Authors’ section and the ‘Reasons To Reject’ section sequentially.
>
> **[Q1: Positional Encoding and BiLSTM]**
>
> Thank you for posing such an insightful question. The main purpose of using the BiLSTM layer and Multi-Head Attention blocks sequentially was to encode both contextual flow and discourse-level relations. We do agree that the positional encoding in conventional Transformers is made for a similar reason. However, positional encoding only indicates the sequential order of the sentences. While a lot of news articles typically follow an inverse pyramid structure, it does not mean that the sentences given in the news article are sequentially related to each other. For example, the lead of a news article usually contains a high-level description of the topic but the supporting points might stay far from each other. This pattern is a common occurrence in lengthy news articles. Our purpose in adopting BiLSTM was to capture these intricate contextual relationships between sentences that are more complex than simple sequential ordering that positional will encode. This finding can also be supported by established analysis on the limitation of positional encoding (Language Models with Transformers (Wang et al., 2019)).
>
>
> **[Q2: Suggestion for Figure 3]**
>
> Thank you for the suggestion, we can work to improve the clarity of the diagram.  The intent of this diagram is to show the comparison in learning curves between BERT and our model. We will make this graph more clearly show that our model has lower variance and higher accuracy compared to BERT.
>
>
> **[Q3: Suggestion for Table 3 Caption]**
>
> Thank you for suggesting to make the caption of Table 3 clearer. In the final version of the paper, will change the caption of the table to be more clear. As either lexically or informationally biased sentences tend to contain the writing style of the journalist or the news outlets, we reported the BERTScore between the set of biased sentences and the abstractive summaries generated by BART-Large along with the comparison with our paper. We will reiterate the definitions of ‘lexical bias’ and ‘informational bias’ from Fan et al., 2019 and briefly introduce the insight that we had from the table after modifying the caption.
>
> **[R1: Comparison with Other Models]**
>
> Thank you for highlighting the importance of a comprehensive comparison with models other than BERT. We strongly concur that considering models that are specialized for political bias detection, can be helpful for fair comparison. During our research, however, we had difficulty finding relevant models suitable for comparison.  While there are specialized models for political bias detection like POLITICS (Liu et al., 2022) it is impossible to test them in our dataset setting as they require additional information or pre-processing from the plain text of the news articles: external knowledge (Baly et la., 2020, Zhang et al., 2022, Feng et al., 2022), specifically knowledge triplets (Liu et al., 2022, Kim and Johnson et al, 2022). In contrast, our method emphasizes content-based classification without the use of knowledge augmentation, ensuring easy reproducibility and independence from external knowledge sources. We view it as a significant strength of our model and for its distinctness, it is difficult to find models with a similar methodology for a direct comparison.
> Though we couldn't find directly comparable state-of-the-art models for a fair comparison, we conducted additional experiments to overcome the limitation. In follow-up experiments based on your comments we further found that when compared to other language models of different sizes(e.g., GPT-2, Bert-Large, Roberta-Base, Roberta-Large), our methodology consistently ensures robustness.
>
> We experimented BERT-Large to empirically show that the variance of AUROC and F1 Score should be higher as the number of model’s parameters increase. Also, we experimented GPT2 to empirically show that same phenomena that we focus on also happens in stronger language model with more parameters and more input token limits. The below table shows that GPT2 got almost 10% point of difference in Macro F1 score in both 1,000 and 10,000 instances setting. Also, BERT-Large got more than 7% point difference in Macro F1 score in both 1,000 and 10,000 instances setting. It is notable that bigger and longer model like GPT2 and BERT-Large is still more variant than our model to the training and test sets.
>
>
> |        |            | GPT2                |                     | BERT-LARGE      |                 | Ours                |                     |
> |--------|------------|---------------------|---------------------|-----------------|-----------------|---------------------|---------------------|
> |        |            |        AUROC        |       Macro F1      |      AUROC      | Macro F1        | AUROC               | Macro F1            |
> | 1,000  | Test Set 1 | 0.5050 (0.0260)     | 0.3214 (0.0145)     | 0.5229 (0.0279) | 0.3180 (0.0585) | **0.6017 (0.0190)** | **0.4227 (0.0185)** |
> |        | Test Set 2 | **0.6185 (0.0174)** | **0.4353 (0.0428)** | 0.5858 (0.0532) | 0.3871 (0.0849) | 0.5703 (0.0298)     | 0.3971 (0.0250)     |
> | 10,000 | Test Set 1 | 0.5756 (0.0122)     | 0.3883 (0.0425)     | 0.5596 (0.0158) | 0.3841 (0.0131) | **0.6529 (0.0081)** | **0.4747 (0.0090)** |
> |        | Test Set 2 | 0.6564 (0.0134)     | 0.4833 (0.0269)     | 0.6559 (0.0231) | 0.4912 (0.0183) | **0.6783 (0.0094)** | **0.5089 (0.0122)** |
>
>
>
> **[R2: Statistical Analysis in Section 7.3]**
>
> In the study "We Can Detect Your Bias: Predicting the Political Ideology of News Articles" by Baly et al. (2020), inconsistencies were underscored when classifying political bias in newspaper articles, especially when models are exposed to unfamiliar news outlets. Echoing their findings, our research aimed to create a model that remains consistent in its classifications, irrespective of the outlet's unique writing style. We noticed challenges, marked by fluctuating AUROC scores when testing on test data consists purely of new sources. Our hypothesis posited that our model would demonstrate a tighter standard deviation in AUROC scores compared to the BERT classifier, even when subjected to varied datasets or different training data volumes. ANOVA would have been a good method if we wanted to show that the performance of the model is invariant to the size of the training set. However, we would like to clarify that we intended to share the robustness of the model by comparing BERT and our model with a randomly sampled (by seed) training set in a fixed number of samples. In this manner, we couldn't apply ANOVA to our experiment. We acknowledge that the intention of our experiments lacks understandability, and we will try to explain our intention as clearly as possible.
>
> **[R3: Validity of K-Means Clustering]**
>
> Thank you for your compliment on our discourse analysis in Section 8, and we would like to provide further support for Section 8 including the aggregated score. We ran clustering 5 times by changing seeds, and we got the exact same result in 4 cases. In one case, a single data point originally in Cluster 3 was allocated to Cluster 1. As a result, we got the exact same clustering result with the aggregated score. For further justification, we would like to share some statistics to validify our analysis:
>
> 1) Silhouette Score: Silhouette score (1~-1) with three clusters was 0.8988, and it linearly decreased as the number of clusters increased.
>
> 2) More than Three Clusters: We tried clustering into three to five clusters, and the biggest clusters took 92% (result in our paper, K=3),  91.33% (K=4), and 88% (K=5) of the whole set.
>
> 3) Clustering within Cluster 1: We tried re-clustering the news articles in Cluster 1 into two to five clusters, and the biggest clusters took 97.6% (K=2), 90.9% (K=3), 87.3% (K=4), and 86.2% (K=5) of news articles in Cluster 1.
>
>
> We believe that these results further clarify that the majority of news articles in Cluster 1 share a strong similarity, and Cluster 2 and 3 have distinct features that clearly separate them from the ones in Cluster 1 in terms of the location of the main sentences.
>
>
> **[R4: Clarification in Label Propagation]**
>
> Along with your kind suggestions in Q2 and Q3, thank you again for suggesting the clarification of notations and writings. We will take steps to address the observations you’ve made. For example, we would be happy to clarify the writing in lines 332 to 335 and the notation in Equation 4 in the final manuscript. We will also remind the reader of this in Section 8 when we perform the analysis.

---

### Meta-Review · Area_Chair_GYpg · 2023-09-22

**Recommendation:** 2

**Metareview:**

The authors proposed a new modeling formulation that examines and addresses the domain dependency issue in political bias detection models. The authors tried to remove dependency on external knowledge sources. Reviewers have concerns regarding comparison with baselines.

All the reviewers agree that the paper provides sufficient support for the major claims.

However, the contributions are marginal. Comparisons are not comprehensive (Reviewers Nv3C, twSB). Some terms need to be defined. An additional revision would be helpful to make the paper more readable (Reviewer Nv3C, b9Zn).

---

### Decision · Program_Chairs · 2023-10-07

**Decision:**

Accept-Findings

**Comment:**

The authors proposed a new modeling formulation that examines and addresses the domain dependency issue in political bias detection models. The authors tried to remove dependency on external knowledge sources. Reviewers have concerns regarding comparison with baselines.

All the reviewers agree that the paper provides sufficient support for the major claims.

However, the contributions are marginal. Comparisons are not comprehensive (Reviewers Nv3C, twSB). Some terms need to be defined. An additional revision would be helpful to make the paper more readable (Reviewer Nv3C, b9Zn).